# Facile Synthesis of Formaldehyde-Free Bio-Based Thermoset Resins for Fabrication of Highly Efficient Foams

**DOI:** 10.3390/polym14235140

**Published:** 2022-11-25

**Authors:** Xuehui Li, Bowen Liu, Lulu Zheng, Hisham Essawy, Zhiyan Liu, Can Liu, Xiaojian Zhou, Jun Zhang

**Affiliations:** 1Yunnan Provincial Key Laboratory of Wood Adhesives and Glued Products, Southwest Forestry University, Kunming 650224, China; 2Department of Polymers and Pigments, National Research Centre, Dokki, Cairo 12622, Egypt; 3Graduate School of Agriculture, Kyoto University, Kitashirakawa-oiwake-cho, Sakyo-ku, Kyoto 606-8502, Japan

**Keywords:** starch, tannin, glyoxal, furfuryl alcohol, foam

## Abstract

Bio-based biodegradable foams were formulated from a crosslinkable network structure combining starch, furfuryl alcohol, glyoxal, and condensed tannin in the presence of p-toluenesulfonic acid (pTSA) and azodicarbonamide (AC) as a foaming agent. More importantly, the reinforcement of gelatinized starch–furanic foam using tannin, originating from forestry, resulted in an excellent compressive strength and lower pulverization ratio. Moreover, the addition of tannin guaranteed a low thermal conductivity and moderate flame retardancy. Fourier transform infrared (FTIR) spectroscopy approved the successful polycondensation of these condensing agents under the employed acidic conditions. Moreover, the catalytic effect of pTSA on the foaming agent induced liberation of gases, which are necessary for foam formation during crosslinking. Scanning electron microscopy (SEM) showed foam formation comprising closed cells with uniform cell distribution and appropriate apparent density. Meanwhile, the novel foam exhibited biodegradation under the action of *Penicillium* sp., as identified by the damage of cell walls of this foam over a period of 30 days.

## 1. Introduction

Foam materials can be used as building and packaging materials because of their advantages such as lightweight, heat insulation, flame retardancy, sound insulation, and shock absorption [1,2,3,4]. However, the oil-derived foams on the market, such as polyurethane (PU) [5,6] and polystyrene (PS) [7,8], are nonrenewable materials that exhibit high carbon emission mode dominated by fossil energy. Although PU foams have good thermal insulation, they burn easily to release the highly toxic hydrogen cyanide, which causes great harm to human health and the environment. The mainstream energy of industrial development in most countries is still biased towards fossil energy. This results in a high level of carbon emission, which seriously affects the development of low-carbon environment and energy conservation [9].

Biomass energy is clean, nontoxic, widely sourced, and renewable. It has become the fourth largest source of energy after coal, oil, and natural gas [10,11]. Improving the utilization rate of biomass energy is an effective way to achieve a low-carbon environment.

Biomass materials originating from agriculture and forestry are important components of biomass energy. Among them, the planting area and total output of corn are second only to rice and wheat, thus they are regarded as an important renewable biomass resource from crops [12]. However, corn grains are mainly used as food, while corncobs are usually removed via burning in open air without efficient utilization. A large amount of residues, resulting from complete combustion, cause the release of numerous quantity of SO_2_ and CO_2_ into the atmosphere, which is further polluting the environment [13,14].

Therefore, the use of corn as a raw material for foam production can not only reduce the dependence of the foam industry on petroleum/petrochemical resources but also make full use of corncob and reduce the release of harmful gases during combustion.

Starch, which exists widely in corn grain as a low-cost biodegradable polymer, has been applied in the fields of wood adhesives [15,16,17], thermoplastic foam [18,19,20], and thermoplastics [21,22]. As a thermoplastic foam material, starch has poor heat resistance. Meanwhile, its high viscosity and poor fluidity, which are required for facile processing, result in poor mechanical properties of the derived foam [18,19]. In order to improve the mechanical properties and thermal stability of starch-based foams, some studies have been carried out that involve the use of hemp fiber or nano cellulose to strengthen the resulting foam [23,24], using a blend of beeswax and starch [25], or using a crosslinked structure of chitosan and starch [26]. Although the mechanical properties and thermal stability of thermoplastic foam were improved by modification, the process of modification is complex and tedious. Meanwhile, compared with some thermosetting foams, such as those derived from phenol formaldehyde (PF) resin [27,28], the modified thermoplastic starch foams were lacking good hardness and reasonable thermal stability.

Interestingly, furfuryl alcohol, which is mainly extracted from corncob, has a stable furan ring and reactive hydroxymethyl group. It is often used to prepare bio-based materials with high hardness and good thermal performance, such as thermosetting rigid plastic [29], foams [30,31,32,33], wood adhesives [34,35,36], and grinding wheels [37,38,39]. Although the polycondensation of starch and furfuryl alcohol is considered to yield foams with good thermal stability and hardness, the hydroxymethyl group in furfuryl alcohol can react with the adjacent furfuryl alcohol under acidic conditions [29], so it is easy for it to undergo self-polycondensation under acidic conditions, while it is difficult for it to undergo further crosslinking with starch. It has been reported that pretreatment of furfuryl alcohol with formaldehyde can reduce the self-condensation of furfuryl alcohol [40], even though formaldehyde is volatile and harmful to human health. In our previous study [35], the use of a lower-toxic, nonvolatile glyoxal, instead of formaldehyde, to react with furfuryl alcohol could also reduce the self-condensation of furfuryl alcohol. At the same time, it was found that during the mixing process of starch and furfuryl alcohol, the gelatinized starch could cause high viscosity, which resulted in poor fluidity and difficulty of processing [35], so it was difficult to produce uniformly distributed bubbles. In addition, the prepared foams using condensed tannin and furfuryl alcohol have been carefully studied [33] due to their low viscosity and ease of foaming. Therefore, it is interesting to employ condensed tannins to reduce the viscosity of the starch and furfuryl alcohol resin system. Moreover, diethyl ether is the main foaming agent for preparation of tannin–furanic foams [41,42]; however, its high volatility and rapid foaming make the process difficult to control, especially when its high toxicity is considered. On the other hand, azodicarbonamide (AC) is also a known foaming agent [43,44], and its decomposition products are nontoxic, odorless, and nonpolluting.

Keeping the above arguments in mind, this study will focus on the use of gelatinized starch and furfuryl alcohol from corn as the main raw materials, glyoxal as a crosslinking agent, partially condensed tannin as a reactive co-condensing agent for viscosity control of the system, and AC as a foaming agent for preparation of a formaldehyde-free biomass-based thermosetting foam.

## 2. Materials and Methods

### 2.1. Materials

Mimosa (*Acacia mearnsii* De Willd) tannin extract powder (T) was purchased from the Wuming Grilled Rubber Factory (Guangxi, China). Corn (*Zea mays* L.) starch (S) was provided by Jilin COFCO Biochemical Energy Sales Co., Ltd (Changchun, China). Furfuryl alcohol (FA, with a purity of 98%), formaldehyde (F, with a purity of 37%), glyoxal (G, with a purity of 40%), p-toluenesulfonic acid (pTSA, with a purity of 97.5%), and silicone oil (with a viscosity of 100 mm^2^/s) were obtained from Sinopharm, Beijing, China. Azodicarbonamide (with a purity of 98%) was obtained from Macklin’s Reagent Co., Ltd. (Shanghai, China), *Penicillium* sp. colonies were prepared as follows: Wild *Armillaria mellea* was collected in Daguan, Zhaotong, China. The bacteria was placed in a sealed bag, then wetted at room temperature and cultured for 1–3 days. After the mycelium growth on the bacterial surface, it was inoculated into a sterile medium and cultured for 1–2 days at a temperature of 28 °C and relative humidity of 75% to obtain the *Penicillium* sp. colonies.

### 2.2. Preparation of Tannin–Starch–Glyoxal–Furfuryl Alcohol (TSGFA) Resin-Based Foam

Table 1 gives the detailed formulation for preparation of TSGFA resin-based foam. First, starch, tannin, and furfuryl alcohol were mixed in a beaker using a stirrer (JJ-200 Chengdu Test Instrument Co., Ltd., Chengdu, China) for 4 min, then glyoxal was added and the stirring was continued for 4 min before pTSA (65% aqueous solution) was added slowly to obtain TSGFA resin within few minutes. The other resins, tannin–starch–formaldehyde–furfuryl alcohol (TSFFA), prepared by replacing glyoxal with formaldehyde, starch–glyoxal–furfural alcohol (SGFA), and tannin–starch–furfural alcohol (TSFA), were also prepared for comparison with TSGFA resin.

Subsequently, silicone oil as a release agent and AC as a foaming agent were added into the TSGFA, TSFFA, SGFA, and TSFA resins, respectively, and the corresponding mixtures were homogenized using a simple agitator (HM-955, Dong Ling Electric Co., Ltd., Guangzhou, China) at a speed of 1500 r/min for 10 min. Then, each mixture was poured into a mold with a size of 90 mm × 90 mm × 90 mm. Afterwards, the mold was transferred to an oven (101, Rongshida Electronic Equipment Co., Ltd., Kunshan, China), where curing was achieved at 80 °C for 24 h to obtain TSGFA-, TSFFA-, SGFA-, and TSFA-derived foam samples, respectively. The preparation process of TSGFA-based foam is shown in Figure 1.

### 2.3. Characterizations

The prepared foam samples were placed in a room at 20 °C and 50% relative humidity for 1 day. Subsequently, the performance was evaluated by conducting some tests, in which every test was repeated five times, and the average value was considered.

The structure of foam samples was elucidated using a Varian-1000 infrared spectrometer (Varian, Palo Alto, CA, USA). The foam samples were ground into powder (particle size around 35–38 µm) with a grinder (jms-130a, Jingfu, Guangzhou, China). One g of KBr was mixed with 0.01 g of each foam powder, and the runs were conducted over the wavenumber range of 400–4000 cm^−1^.

The measurements of apparent density were accomplished according to the Chinese national standard GB/T 6343–2009. The apparent density was calculated using Equation (1).
(1)$$\rho =\frac{m}{v}\times 10^{6}$$
where m is the mass of the foam sample in g, v is the volume of the foam sample in mm^3^, and ρ is the apparent density in kg/m^3^.

A scanning electron microscope (s-4160 Fe, Hitachi, Tokyo, Japan) was used for observation of the microstructural details of foam samples with a size of 10 mm × 10 mm × 10 mm.

The cell size and cell wall thickness of the various samples were calculated from the obtained SEM images with the help of Nano Measurer 1.2 software (Microsoft, Redmond, WA, USA).

A universal testing machine (AG-50KN, SHIMADZU, Berlin, Germany) was used to evaluate the compressive strength of foam samples, cut to a size of 30 × 30 × 30 mm^3^, at 25 °C and relative humidity of 45–65% by employing a compression rate of 2 mm/min.

The pulverization of the foam samples was evaluated according to the Chinese national standard GB/T 12812-2006 on samples with a size of 5 cm × 5 cm × 5 cm. The foam samples were placed horizontally on sandpaper (400 mesh) with a length of 250 mm, while 200 g iron prop was placed on the foam. The foam sample was pulled from one section of the sandpaper to the other end, and the pulling speed was ensured to be the same every time. When a repetition of 30 times was completed, the remaining quantity of each foam sample was recorded by employing Equation (2).
(2)$$M=\frac{m_{0}-m_{1}}{m_{0}}\times 100\%$$
where M is the pulverization rate, %; m_0_ is the initial weight of the foam, g; m_1_ is the weight of the foam after being pulverized by the sandpaper, g.

The tensile parameters of the foam were examined according to the Chinese national standard GB/T528-2009, on samples with a size of 30 × 30 × 15 mm^3^, while the load rate was set at 1 mm/min.

A thermal conductivity meter (Ybf-2, Dahua Technology Co., Ltd., Hangzhou, China) was used to measure the thermal conductivity on foam samples, customized into a cylinder shape with a radius (R) of 50 mm and 10 mm height (h), according to Equation (3):(3)$$\lambda =-\mathrm{mc}\frac{2h_{p}+R_{p}}{2h_{p}+2R_{p}}\times \frac{1}{{\pi R}^{2}}\times \frac{h}{T_{1}-T_{2}}\times \frac{\mathrm{dT}}{\mathrm{dt}}|T=T_{2}$$
where λ is the thermal conductivity, W·m^−1^·K^−1^; m is the mass of the lower copper plate, g; c is the specific heat capacity of the bottom copper plate of the instrument, $R_{p}$ and $h_{p}$ are the radius and thickness of the lower copper plate, mm; R is the radius of the foam sample, mm; h is the height of the foam sample, mm; $T_{1}-T_{2}$ is the temperature difference between the upper and lower copper plates; $\frac{\mathrm{dT}}{\mathrm{dt}}|T=T_{2}$ is the cooling rate of copper plate exposed to air.

Thermogravimetric analyzer (TG 209 F3, Netzsch, Selb, Germany) was used to investigate the thermal degradation behavior of the foam samples, where a heating rate of 20 °C/min was employed under nitrogen atmosphere over the temperature range 30 to 800 °C.

*Penicillium* sp. was used to check the biodegradability of the TSGFA-based foam sample according to other reported studies [45,46]. The TSGFA-based foam sample was placed in a Petri dish, inoculated with *Penicillium* sp., then the Petri dish was covered with a parafilm (pm996, Bemis, Neenah, WI, USA). Then, it was kept at 28 °C and 75% relative humidity for 30 days. Eventually, the weight change of the TSGFA foam sample after the action of *Penicillium* sp. was recorded and compared to the initial weight. After that, the TSGFA foam treated by *Penicillium* sp. until the 30th day (a size of 3 mm × 3 mm × 3 mm) was fixed using 2.5% glutaraldehyde at 4 °C for 12 h. The growth and action of *Penicillium* sp. colonies on the TSGFA foam sample were additionally studied using a field emission scanning electron microscope (FESEM), Hitachi su8010 (Hitachi, Ltd., Tokyo, Japan).

## 3. Results and Discussion

Figure 1 shows the FTIR spectra of S, T, SGFA, and TSGFA foams. The stretching vibration of -OH appeared at 3379–3449 cm^−1^. However, due to the induction effect of different side groups, the characteristic absorption peaks of -OH in SGFA and TSGFA foams seemed different from those of S and T; the peak at 1720 cm^−1^ represents the skeleton vibration of aromatic hydrocarbons. The hydroxymethyl of furfuryl alcohol reacts with C6 or C8 in the A-ring of tannin, the P-Π conjugation effect makes the skeleton vibration of aromatic hydrocarbons more intense, so that the peak at 1720 cm^−1^ became more obvious in the case of TSGFA. The peaks in the range 1652–1613 cm^−1^ are attributed to the C–H stretching vibration of all polymeric and small structural units. Due to the influence of the side groups connected to the aromatic rings, the C–H absorption peak of TSGFA foam was different from that of S in terms of intensity and position because the conjugation effect of the aromatic rings caused the C–H absorption peak of TSGFA foam to undergo a shift. The absorption peak at 1567 cm^−1^ in the case of TSGFA foam, which is due to the etherification of the tertiary carbon atom of the side group of furfuryl alcohol, looks dissimilar in the case of SGFA, T, and S, indicating formation of condensation products from reaction of tannin with furfuryl alcohol. The peaks at 1518–1455 cm^−1^, representing the aromatic skeleton, are absent in the relevant spectrum of S, while the peaks at 1283–1125 cm^−1^ are due to C–O absorption of all polymeric, oligomeric, and basic structural units. Furthermore, the peak at 1006 cm^−1^ refers to the C–OH of the furfuryl alcohol high oligomers. The peak at 747 cm^−1^ is indicative of the C–C stretching vibration. Due to the induction effect of different structural units connected by covalent bonds in the case of TSGFA, the characteristic absorption peak of C–C, appearing originally at 747 cm^−1^, shifted to 795 cm^−1^. It can be now recognized that the condensation reactions between starch, tannin, furfuryl alcohol, and glyoxal occurred successfully. The main reactions of the TSGFA foam system are shown in Figure 2a,b.

SEM images, apparent density, and cell size distribution of the different prepared foams are shown in Figure 2. All foams have closed cell structure, and the cells are either round or oval-shaped. This indicates that the AC foaming agent decomposes by the action of pTSA to produce carbon dioxide, nitrogen, and other gases that diffuse evenly during the resin solidification to generate homogeneous foams. However, the average cell size and cell wall thickness of the different foam samples are varied. The average cell wall thickness and apparent density of the TSGFA-based foam sample are greater than those of SGFA and TSFA foams, indicating that the addition of tannin and glyoxal underwent polycondensation reactions with starch and furfuryl alcohol, which increased the integrity of the resin and further upgraded the network structure of the foam system. At the same time, TSGFA- and TSFFA-based foam samples acquired similar average cell size, but the TSFFA-based foam showed smaller apparent density with respect to that of TSGFA-based foam, indicating that as a crosslinking agent, the activity of formaldehyde is higher than that of glyoxal [35], which makes the reaction of tannin, starch, and furfural more efficient, the number of oligomers in the case of TSFFA resin system less, the compatibility of the foaming agent and TSFFA resin better with respect to TSGFA, and the foaming process more uniform, while more bubbles are generated. According to SEM images, the cell shape of TSFFA foam is uniform compared with TSGFA, almost circular. Therefore, the apparent density of TSFFA-based foam is lower with respect to TSGFA.

Figure 3 shows the pulverization ratio and tensile strength of the different starch-based foams, which reveals the extent of damage on the foam when it is cut [41]. The lower the pulverization ratio, the less likely it is to undergo damage. The pulverization ratio of the SGFA foam sample (1.6%) is lower than that of TSGFA (2.1%) and TSFFA (1.7%), which indicates that the addition of tannin can easily improve the pulverization resistance of the foam. This depends on the high hardness acquired by the tannin resin after curing [29].

Meanwhile, according to the tensile strength data of the different foams, the tensile strength of the SGFA-based foam is higher than that of the TSGFA foam sample. Therefore, the addition of tannin improves the foam hardness; however, the toughness of the foam is reduced. At the same time, the pulverization ratio of the TSFFA foam is lower than that of the TSGFA foam. On the contrary, the tensile strength of the TSFFA foam is higher than that of the TSGFA foam, which is accounted for by the higher reactivity of formaldehyde with respect to glyoxal to build up a stronger network structure. However, glyoxal, as a crosslinking agent, improved the crosslinking between tannin, starch, and furfuryl alcohol system, and further reduced the pulverization ratio of the foam, which reached 4.3% in the case of TSFA-based foam. More importantly, compared with some biomass foams, such as tannin–furanic–soybean protein isolate (SPI)-based foam (3.68%) [47] and tannin–formaldehyde–furanic foam (16.49%) [48], the TSGFA-based foam exhibited a lower pulverization ratio.

Figure 4 shows the stress–strain curves of the different prepared foams. It can be seen that the compressive strength at yield of TSGFA-based foam (1.751 MPa) is higher than that of SGFA-based foam (1.486 MPa), which corroborates that addition of tannin improves the compressive strength of the foam, and this depends on the phenolic ring structure of tannin which provides higher hardness [29]. At the same time, the strength at yield of TSFFA-based foam (2.311 MPa) is higher with respect to that of TSGFA-based foam, which is attributed to the higher reactivity of formaldehyde in comparison of glyoxal. It can be also seen from Figure 2 that the cell wall of TSFFA-based foam is thicker than that of TSGFA-based foam, which is consistent with the results of pulverization ratio. At the same time, the compressive strength at yield of TSFA-based foam (1.183 MPa) is lower with respect to that of TSGFA-based foam, indicating the role of glyoxal to promote the condensation reaction of tannin, starch, and furfuryl alcohol. In addition, the compressive strength at yield of TSGFA-based foam is much higher than that of the tannin–formaldehyde–furanic foam (0.18 MPa) and tannin–furanic–soybean protein isolate (SPI)-based foam (0.5 MPa) [47,48]. Thus, bio-based foam structures prepared by combining tannin and starch as condensing agents in the presence of glyoxal as crosslinking agent and AC as foaming agent present potential for more applications compared with foams based exclusively on tannin. It is worthy to note that the strain at break of the TSFFA-based foam is not affected even with the elevation of maximum strength.

Thermal conductivity is an important index to check the efficiency of thermal insulation for a foam material. Figure 5 shows the thermal conductivity of the different prepared foams, which are all characterized by closed cell structure. Therefore, compared with some commercial foams, such as polyethylene foam (0.047 W·m^−1^·K^−1^) [9], the starch-based foam acquired lower thermal conductivity. More importantly, although the average cell size of TSGFA-based foam (210 μm) is higher than that of SGFA foam (200 μm) (Figure 2), the addition of glyoxal improves the crosslinking of the foam system and produces a more uniform closed cell distribution, which leads to lower thermal conductivity of TSGFA-based foam (0.030 W·m^−1^·K^−1^) compared with that of TSFA (0.035 W·m^−1^·K^−1^). Meanwhile, the thermal conductivity of TSGFA-based foam, prepared using glyoxal instead of formaldehyde, is similar to that of TSFFA, which is prepared using formaldehyde. These results show that the TSGFA-based foam has considerable mechanical strength and thermal insulation, which expands its application prospects.

Figure 6 displays the TG–DTG curves of the different foam structures prepared in this study. The mass of the foam decreases slightly between 100 and 280 °C, which reveals humidity and the gas generated by residual blowing agent (AC) when it interacts with pTSA at higher temperature. Figure 6b indicates that with the increase of temperature, the highest degradation rates were accomplished at 200–280 and 420–470 °C, respectively. At the temperature range of 200–280 °C, the starch in the foam starts to degrade, whereas the blowing agent decomposes into carbon dioxide and nitrogen. Moreover, at the range of 420–470 °C, the starch oligomers begin to degrade further. It is obvious that the mass loss in the case of SGFA-based foam is higher with respect to TSGFA-based foam, which indicates that the tannin addition improves the heat resistance of the foam. In addition, the mass loss in the case of TSFA-based foam is larger because no glyoxal or formaldehyde is added, considering that the reactivity of tannin or starch is poor with furfuryl alcohol in the absence of any of these aldehydes. It can be also seen from the curves that TSGFA- and TSFFA-based foams behaved almost the same over the range of 300–800 °C, which illustrates their similar heat resistance.

A butane spray gun was used for characterization of the combustion performance of TSGFA-, TSFFA-, SGFA-, and TSFA-based foams, and the results are presented in Figure 7. At the beginning of the test, the four foam samples were all located at the same position at the nozzle of the spray gun to reach the same combustion temperature. After 65 s, TSGFA- and TSFFA-based foams were partially burned to red (Figure 7c,g). After cooling for a certain time, a small part in both cases was broken with the flame-contacted part, while the untouched part remained intact (Figure 7d,h). However, after 65 s, the SGFA and TSFA foam samples could be observed to be ignited (Figure 7k,o). After cooling for a certain time, both samples were completely broken and carbonized. It can be explained that the addition of tannin and glyoxal to the starch–furfuryl alcohol resin system could make the resin build a dense network structure, which enhanced the flame retardancy of the material by delaying the flame diffusion.

Figure 8 shows the effect of *Penicillium* sp. on the TSGFA-based foam. *Penicillium* sp. has a marked degradative effect on polysaccharides [49]. It can be seen from Figure 8a that at the beginning, *Penicillium* sp. (marked by a rectangle) did not contact the foam sample. On the tenth day, *Penicillium* sp. (marked by an arrow) had surrounded the TSGFA foam. After 20 days, *Penicillium* sp. (marked by an arrow) had run into the TSGFA foam. By completion of 30 days, the foam was covered by *Penicillium* sp. (marked by arrow). The mass loss of materials is the most commonly used method to follow the degradation induced on materials by fungi [50]. Figure 8b shows the mass loss of TSGFA-based foam induced by *Penicillium* sp. The mass loss of the foam was only 0.24% on the 10th day. With the elapse of the time, the mass loss reached 0.68% by the 30th day. In order to further confirm the biodegradation of TSGFA-based foam by *Penicillium* sp., the treated foam with *Penicillium* sp. for 30 days was examined using FESEM. The results are shown in Figure 9a–c, which reveal a large number of *Penicillium* sp. colonies (marked by arrow) diffusing into the TSGFA sample. More importantly, some cell walls of TSGFA-based foam were damaged (marked by the square). Further, it is clear from Figure 9d that the cell wall of TSGFA-based foam was perforated by some *Penicillium* sp. colonies (marked by the arrow). These results present strong proof that TSGFA foam is liable to undergo biodegradation.

## 4. Conclusions


Different bio-based foam structures in crosslinked form can be prepared from a polycondensation reaction incorporating starch, furfuryl alcohol, glyoxal, and condensed tannin under mild acidic conditions and an appropriate foaming agent. The selection of catalytic system and foaming agent determines, to a large extent, the cell formation characteristics of the foam structure, while the mechanical strength is a parameter that is dependent on the condensing agents’ formulation.It can be concluded that the addition of tannin into the formulation contributed significantly to the high compressive strength and low pulverization ratio.The crosslinking between tannin, starch, glyoxal, and furfuryl alcohol under the employed reaction conditions provoked formation of closed cells with uniform cell distribution and appropriate apparent density, which contributed to the good thermal insulation and flame retardancy of the foam.The induced biodegradability of the prepared foams using *Penicillium* sp. is ascribed to the bio-based nature of the structural units involved in building the chemical skeleton of the foam.


## Data Availability

Not applicable.

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
