# Peer review of "Facile Synthesis of Formaldehyde-Free Bio-Based Thermoset Resins for Fabrication of Highly Efficient Foams"

_polymers, 2022, doi:10.3390/polym14235140_

Round 1

Reviewer 1 Report

Manuscript Number: Polymers-2041877

The manuscript written by Li et al. titled “Facile Synthesis of Formaldehyde-Free Bio-Based Thermoset Resins for Fabrication of Highly Efficient Foams” studied the effects of glyoxal and condensed tannin as crosslinker in place of formaldehyde to produce biobased and degradable stable foam. After careful reading the paper, I agree for the publication of the manuscript in the Polymers and want to suggest a few comment-

1.      It would be preferable to use different abbreviation for the furfuryl alcohol and formaldehyde.

2.       The axis titles in the figures can be reduced.

3.      For the foam SEM images please refer to are they topical or cross-sectional. If they are topical, please also provide the cross-sectional images.

4.      Please check the other grammar mistakes and please limit the self-citation to 10% unless very important.

Author Response

Dear Reviewer:

Thanks for your letter and the constructive comments of the reviewer concerning our Manuscript. ID:polymers-2041877 The comments are really valuable and very helpful for improving our manuscript, and can significantly guide our research in the future. We have studied these comments carefully and made all the required modifications, which we hope to meet your approval. The main corrections in the manuscript and the responses to the reviewer’ comments are listed below.

The manuscript written by Li et al. titled “Facile Synthesis of Formaldehyde-Free Bio-Based Thermoset Resins for Fabrication of Highly Efficient Foams” studied the effects of glyoxal and condensed tannin as crosslinker in place of formaldehyde to produce biobased and degradable stable foam. After careful reading the paper, I agree for the publication of the manuscript in the Polymers and want to suggest a few comments

  1. It would be preferable to use different abbreviation for the furfuryl alcohol and formaldehyde.

Reply to reviewer: We have made changes in the article and abbreviated formaldehyde as F and furfuryl alcohol as FA, and the changes are marked with yellow highlighting.

  1. The axis titles in the figures can be reduced.

Reply to reviewer: We reduced the axis titles in Figures 3 and 5.

  1. For the foam SEM images please refer to are they topical or cross-sectional. If they are topical, please also provide the cross-sectional images.

Reply to reviewer: The SEM images of the foam are cross-sectional images (not topical), we marked that in the article with yellow highlighting.

  1. Please check the other grammar mistakes and please limit the self-citation to 10% unless very important.

Reply to reviewer: Thanks very much for the comments of the reviewers. We have carefully revised the grammar. Regarding for the self-citation, our research group has been involved on research of tannin resin based adhesives, grinding wheels and foams, and these previous studies have provided an important basis for the development of this new tannin-starch based foam. Therefore, we have quoted previously published literatures, which is very necessary.

Reviewer 2 Report

The manuscript titled “Facile Synthesis of Formaldehyde-Free Bio-Based Thermoset 1 Resins for Fabrication of Highly Efficient Foams” proved that bio-based foams with starch, furfuryl alcohol, glyoxal and condensed tannin (TSGF) prepared by using p-toluenesulfonic acid (pTSA) and azodicarbonamide (AC) as a foaming agent showed a low pulverization ratio, good thermal insulation, and biodegradation. Compared with TSFF, the TSGF exhibited equivalent mechanical strength and thermal insulation, while the glyoxal used in TSGF showed lower toxic and nonvolatile. It can be accepted for publication after some revisions detailed below.

1. Table 1:Why is the sample of TSF listed in this study? What is the purpose? Based on the introduction, TSFF was listed to compare the difference between glyoxal and formaldehyde, and SFG was listed to show the effect of tannin.

2. Page 6, line 217-222: “At the same time, TSGF and TSFF foam samples acquired similar average cell size... and the distribution of cell more uniform.” Please explain that the enhanced activity of formaldehyde as a cross-linking agent would lead to a smaller apparent density.

3. Page 6, line 231-232: “Therefore, the addition of tannin improves the hardness of foam, but reduces the toughness of the foam.” No virtual data and references to prove that the introduction of tannin could reduce the toughness of the foam.

Author Response

Dear Reviewer:

Thanks for your letter and the constructive comments of the reviewer concerning our Manuscript. ID:polymers-2041877 The comments are really valuable and very helpful for improving our manuscript, and can significantly guide our research in the future. We have studied these comments carefully and made all the required modifications, which we hope to meet your approval. The main corrections in the manuscript and the responses to the reviewer’ comments are listed below.

The manuscript titled “Facile Synthesis of Formaldehyde-Free Bio-Based Thermoset  Resins for Fabrication of Highly Efficient Foams” proved that bio-based foams with starch, furfuryl alcohol, glyoxal and condensed tannin (TSGF) prepared by using p-toluenesulfonic acid (pTSA) and azodicarbonamide (AC) as a foaming agent showed a low pulverization ratio, good thermal insulation, and biodegradation. Compared with TSFF, the TSGF exhibited equivalent mechanical strength and thermal insulation, while the glyoxal used in TSGF showed lower toxic and nonvolatile. It can be accepted for publication after some revisions detailed below.

  1. Table 1: Why is the sample of TSF listed in this study? What is the purpose? Based on the introduction, TSFF was listed to compare the difference between glyoxal and formaldehyde, and SFG was listed to show the effect of tannin.

 Reply to reviewer: Thanks for the comments of reviewers, we made TSF foam because in the previous study, the hydroxymethyl group of furfuryl alcohol would react with C6 and C8 positions of tannin A ring under acidic conditions, so we wanted to let the readers know whether furfuryl alcohol can directly react with tannin or starch to prepare foam without the crosslinking effect of glyoxal and formaldehyde, so we made TSF samples for comparison. The F denotes furfuryl alcohol, not formaldehyde. In order to distinguish between the abbreviations of furfuryl alcohol and formaldehyde, we labeled the abbreviations of formaldehyde as F and furfuryl alcohol as FA, and marked that with yellow highlighting.

  1. Page 6, line 217-222: “At the same time, TSGF and TSFF foam samples acquired similar average cell size... and the distribution of cell more uniform.” Please explain that the enhanced activity of formaldehyde as a cross-linking agent would lead to a smaller apparent density.

 Reply to reviewer: Thank the reviewers for their comments. We explained that in the article. The number of oligomers in the TSFFA resin system is less, the compatibility of foaming agent and TSFFA resin is better than that of TSGFA, the foaming is more uniform, and more bubbles are generated. According to SEM, the cell shape of sample TSFFA is uniform compared with TSGFA, almost circular. Therefore, the apparent density of sample TSFFA is lower than TSGFA. We marked that with yellow highlighting.

  1. Page 6, line 231-232: “Therefore, the addition of tannin improves the hardness of foam, but reduces the toughness of the foam.” No virtual data and references to prove that the introduction of tannin could reduce the toughness of the foam.

Reply to reviewer: Thanks for the comments of reviewers, we supplemented the tensile strength data of the foam, and the results were included in the article, as shown in Fig. 3., which shows obviously that the addition of tannin reduced the toughness of foam to a certain extent, We also marked this with yellow highlighting.

Reviewer 3 Report

1.      Abstract section require rewrite and give major finding (numerical data)

2.      Give the novelty of the study

3.      What is purpose of this study

4.      What is application in medical other filed

5.      what is application in agricultural filed

6.      Conclusion not explain the finding  

Author Response

Dear Reviewer:

Thanks for your letter and the constructive comments of the reviewer concerning our Manuscript. ID:polymers-2041877 The comments are really valuable and very helpful for improving our manuscript, and can significantly guide our research in the future. We have studied these comments carefully and made all the required modifications, which we hope to meet your approval. The main corrections in the manuscript and the responses to the reviewer’ comments are listed below.

  1. Abstract section require rewrite and give major finding (numerical data)

Reply to reviewer: The abstract has been rewritten to show our best results, and we marked it with yellow highlighting.

  1. Give the novelty of the study

Reply to reviewer: The novelty of this study lies in the use of starch and furfuryl alcohol, both are from corn, for co-condensation, and additional use of condensed tannin from bark, as a reinforcing agent to prepare thermosetting biodegradable foam with high compressive strength, heat insulation and flame retardancy. This foam has not been reported before and is expected to present a good application prospect.

  1. What is purpose of this study

Reply to reviewer: The starch foam studied previously is mainly thermoplastic foam, which has poor thermal stability and low density, leading to weak mechanical properties of foam. Therefore, the purpose of this study is to prepare biodegradable thermosetting starch-based foam with better mechanical properties and thermal stability.

  1. What is application in medical other filed

Reply to reviewer: The foam prepared can be used as building materials, as thermal insulation materials inside walls, or as thermal insulation materials for wood structures of building walls. At present, we have not thought yet of applying it to the medical field.

  1. What is application in agricultural filed

Reply to reviewer: Thank the reviewers for their comments. We use raw materials from agriculture and forestry to prepare the biodegradable foam, at present, we have not thought yet of applying it to the agricultural field.

  1. Conclusion not explain the finding  

Reply to reviewer: The conclusion has been rewritten to show our best results, marked with yellow highlighting.

Reviewer 4 Report

In this contribution, the authors prepared bio-based foams. The effects of monomers on the morphology, mechanical performance, thermal conductivity and stability of the resulting foams were investigated. Although this topic is inspiring to the readership of Polymer, this research overlooked some critical information/discussion, as listed below.

1. While the disadvantage of burning easily is mentioned for PU foams, how is the flame retardancy of the TSGF foam in this research? For example, what are the extent and time of burning?

2. In Line 49 to 51, what is the specific bio-sourced raw material for the TSGF foam, corn or corncob? It is important to distinguish this concept, because corn is the world's most important staple food, whereas corncob is a byproduct.

3. Dose TSF foam have glyoxal? Line 115 does not include glyoxal for TSF, but Table 1 includes glyoxal for TSF. 

4. In Figure 1, what do the FTIR spectra near 1800 cm-1 look like? Some important chemical bonds show peaks near 1800 cm-1, e.g., carbonyl. In addition, the caption defines SGF and TSGF as the same composition. At least, the chemical structure of TSGF foam needs to be illustrated to show how different monomers form chemical bonds with each other. This is not only important to interpret the FTIR spectra, but may also explain why foams based on different compositions show different pulverization ratios and compressive strength.

5. The scale bars in Figure 2 are blurry. Additionally, besides the macropores with a width greater than 100 microns, the SEM images also show much smaller pores incorporated in cell walls. Is the pore size distribution of those smaller pores characterized, e.g., BET? Frequently, the smaller pores are important for a high specific surface area, therefore, impacting the sorption behavior.

6. The weight loss due to the biodegradation is subtle, i.e., less than 1% after 30 days. However, do the compositional or mechanical properties change, e.g., FTIR spectra or compressive strength?

7. Other comments:

Line 47, is SO2 a product from incomplete or complete combustion?

Line 85, does "diethylene ether" refer to diethyl ether?

Author Response

Dear Reviewer:

Thanks for your letter and the constructive comments of the reviewer concerning our Manuscript. ID:polymers-2041877 The comments are really valuable and very helpful for improving our manuscript, and can significantly guide our research in the future. We have studied these comments carefully and made all the required modifications, which we hope to meet your approval. The main corrections in the manuscript and the responses to the reviewer’ comments are listed below.

In this contribution, the authors prepared bio-based foams. The effects of monomers on the morphology, mechanical performance, thermal conductivity and stability of the resulting foams were investigated. Although this topic is inspiring to the readership of Polymer, this research overlooked some critical information/discussion, as listed below.

  1. While the disadvantage of burning easily is mentioned for PU foams, how is the flame retardancy of the TSGF foam in this research? For example, what are the extent and time of burning?

 Reply to reviewer: Thanks for the reviewer for these comments, we supplemented the combustion experiment, and the results showed that TSGFA foam sample remained mostly intact after burning for 1 minute, indicating that it had a certain flame retardancy. See Figure 7 for specific results and analysis, marked with yellow highlighting.

  1. In Line 49 to 51, what is the specific bio-sourced raw material for the TSGF foam, corn or corncob? It is important to distinguish this concept, because corn is the world's most important staple food, whereas corncob is a byproduct.

 Reply to reviewer: Starch comes from corn grain (we use yellow highlighting marks in the introduction part), and furfuryl alcohol comes from corncob, so the main raw materials of TSGFA foam prepared with starch and furfuryl alcohol come from corn (including corn grain and corncob).

  1. Dose TSF foam have glyoxal? Line 115 does not include glyoxal for TSF, but Table 1 includes glyoxal for TSF. 

Reply to reviewer: We thank the reviewer for the careful check. There is no glyoxal in TSFA. It is a typing mistake in Table 1, and corrected it.

  1. In Figure 1, what do the FTIR spectra near 1800 cm-1 look like? Some important chemical bonds show peaks near 1800 cm-1, e.g., carbonyl. In addition, the caption defines SGF and TSGF as the same composition. At least, the chemical structure of TSGF foam needs to be illustrated to show how different monomers form chemical bonds with each other. This is not only important to interpret the FTIR spectra, but may also explain why foams based on different compositions show different pulverization ratios and compressive strength.

Reply to reviewer: Thanks for these comments. We have added the omitted part of the curve and added the main reactions and chemical structures into TSFGA foam system in this article. See Scheme 2 for details.

  1. The scale bars in Figure 2 are blurry. Additionally, besides the macropores with a width greater than 100 microns, the SEM images also show much smaller pores incorporated in cell walls. Is the pore size distribution of those smaller pores characterized, e.g., BET? Frequently, the smaller pores are important for a high specific surface area, therefore, impacting the sorption behavior.

Reply to reviewer: We modified the SEM images to make the scale bars more clear. For the foam to be used in the adsorption field, BET is indeed a very important parameter, but our foam is mainly used as a structural material because of its high compressive strength. We wanted mainly apply it as building insulation materials, so the research does not focus in moment on the micro pores but on the cell wall, however, we will consider this when expanding the application range further in the future.

  1. The weight loss due to the biodegradation is subtle, i.e., less than 1% after 30 days. However, do the compositional or mechanical properties change, e.g., FTIR spectra or compressive strength?

Reply to reviewer: The question raised by the reviewers is really very good, which is also a problem we have been concerned about. We also tested its mechanical properties, and there was almost no significant change after 30 days because the weight loss is still limited at this stage. At the beginning, we thought that the cross-linked foam did not show biodegradable properties, but after reviewing some literature, we found that a cross-linked polymer can still exhibit biodegradability, but the degradation is relatively slow and extended over prolonged time. Therefore, we will also observe that over a longer time, then we will follow any arising changes of the chemical structure or different properties.

  1. Other comments:

Line 47, is SO2 a product from incomplete or complete combustion?

Reply to reviewer: We have consulted the relevant literature, and it is produced by complete combustion. I changed it in the article.

Line 85, does "diethylene ether" refer to diethyl ether?

Reply to reviewer: Thanks for the reviewer's correction. It is diethyl ether, not diethylene ether. We changed it and marked with yellow highlighting.

Round 2

Reviewer 2 Report

The authors have revised the paper and could be accepted in the current version.

Reviewer 3 Report

Accept 

Reviewer 4 Report

All my questions and comments are sufficiently addressed.